# AmbientGAN: Generative models from lossy measurements

**Ashish Bora**
Department of Computer Science
University of Texas at Austin
`ashish.bora@utexas.edu`

**Eric Price**
Department of Computer Science
University of Texas at Austin
`ecprice@cs.utexas.edu`

**Alexandros G. Dimakis**
Department of Electrical and Computer Engineering
University of Texas at Austin
`dimakis@austin.utexas.edu`

## Abstract

Generative models provide a way to model structure in complex distributions and have been shown to be useful for many tasks of practical interest. However, current techniques for training generative models require access to fully-observed samples. In many settings, it is expensive or even impossible to obtain fully-observed samples, but economical to obtain partial, noisy observations. We consider the task of learning an implicit generative model given only lossy measurements of samples from the distribution of interest. We show that the true underlying distribution can be provably recovered even in the presence of per-sample information loss for a class of measurement models. Based on this, we propose a new method of training Generative Adversarial Networks (GANs) which we call AmbientGAN. On three benchmark datasets, and for various measurement models, we demonstrate substantial qualitative and quantitative improvements. Generative models trained with our method can obtain 2-4x higher inception scores than the baselines.

## 1 Introduction

Generative models are powerful tools to concisely represent the structure in large datasets. An implicit generative model is a mechanism that only specifies a stochastic procedure to produce samples from a probability distribution. These models are attractive since they do not require an explicit parametrization of the probability distribution they are trying to model.

Recently, there has been substantial progress in neural-network based implicit generative models within the autoregressive and the adversarial framework. The adversarial framework was pioneered by Generative Adversarial Networks (GANs) [Goodfellow et al. (2014)]. In these models, a generator network attempts to map samples from a simple low-dimensional distribution (such as standard Gaussian) to points in a high-dimensional space that resemble the learned data distribution. At the same time, a discriminator network attempts to distinguish between real and generated samples. By setting up a min-max game between them, the two networks are jointly trained. The latent probability distribution along with the learned generator network define a stochastic procedure that can produce new samples. The adversarial framework has been shown to be extremely successful in modeling complex distributions [Berthelot et al. (2017); Vondrick et al. (2016); Pascual et al. (2017); Wu et al. (2016)], and the priors induced by these models are useful for various applications [Shrivastava et al. (2016); Ho & Ermon (2016)].

This procedure for training generative models requires access to a large number of fully-observed samples from the desired distribution. Unfortunately, obtaining multiple high-resolution samples can be expensive or impractical for some applications. For example, many sensing and tomography problems (*e.g.* MRI, CT Scan) require a large number of projections for good reconstruction. Compressed sensing [Donoho (2006); Candes et al. (2006)] attempts to ameliorate this problem using

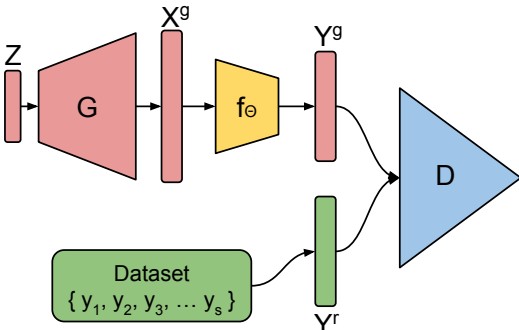

Figure 1: AmbientGAN training. The output of the generator is passed through a simulated random measurement function $f_\Theta$. The discriminator must decide if a measurement is real or generated.

models of the data structure. Recent work has shown that generative models can be particularly effective for easier sensing [Bora et al. (2017); Mardani et al. (2017)]—but if sensing is expensive in the first place, how can we collect enough data to train a generative model to start with?

This work solves this chicken-and-egg problem by training a generative model directly from noisy or incomplete samples. We show that our observations can be even projections or more general measurements of different types and the unknown distribution is still provably recoverable. A critical assumption for our framework and theory to work is that the measurement process is known and satisfies certain technical conditions.

We present several measurement processes for which it is possible to learn a generative model from a dataset of measured samples, both in theory and in practice. Our approach uses a new way of training GANs, which we call AmbientGAN. The idea is simple: rather than distinguish a real image from a generated image as in a traditional GAN, our discriminator must distinguish a real measurement from a simulated measurement of a generated image; see Figure 1. We empirically demonstrate the effectiveness of our approach on three datasets and a variety of measurement models. Our method is able to construct good generative models from extremely noisy observations and even from low dimensional projections with drastic per-sample information loss. We show this qualitatively by exhibiting samples with good visual quality, and quantitatively by comparing inception scores [Salimans et al. (2016)] to baseline methods.

**Theoretical results.** We first consider measurements that are *noisy, blurred* versions of the desired images. That is, we consider convolving the original image with a Gaussian kernel and adding independent Gaussian noise to each pixel (our actual theorem applies to more general kernels and noise distributions). Because of the noise, this process is not invertible for a single image. However, we show that the distribution of measured images uniquely determines the distribution of original images. This implies that a pure Nash equilibrium for the GAN game must find a generative model that matches the true distribution. We show similar results for a *dropout* measurement model, where each pixel is set to zero with some probability $p$, and a *random projection* measurement model, where we observe the inner product of the image with a random Gaussian vector.

**Empirical results.** Our empirical work also considers measurement models for which we do not have provable guarantees. We present results on some of our models now and defer the full exploration to Section 8.

In Fig. 2, we consider the celebA dataset of celebrity faces [Liu et al. (2015)] under randomly placed occlusions, where a randomly placed square containing $1/4$ of the pixels is set to zero. It is hard to inpaint individual images, so cleaning up the data by inpainting and then learning a GAN on the result yields significant artifacts. By incorporating the measurement process into the GAN training, we can produce much better samples.

In Fig. 3a we consider learning from noisy, blurred version of images from the celebA dataset. Each image is convolved with a Gaussian kernel and then IID Gaussian noise is added to each pixel.

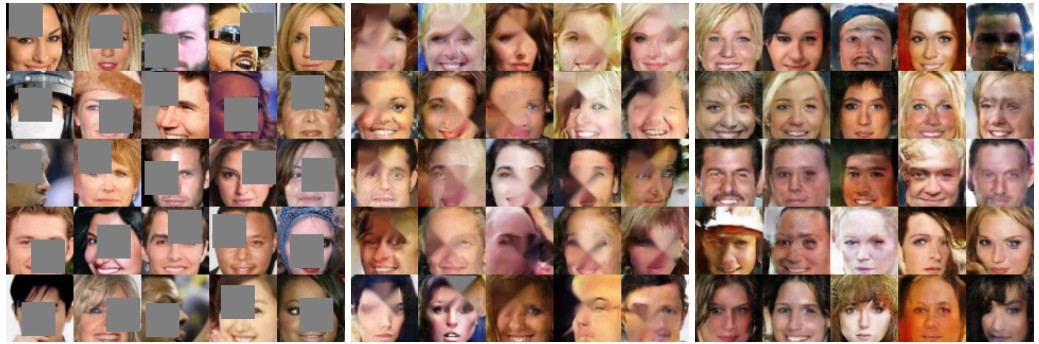

Figure 2: (Left) Samples of lossy measurements used for training. Samples produced by (middle) a baseline that trains from inpainted images, and (right) our model.

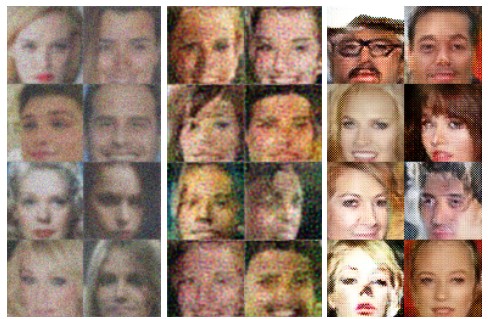

(a) (left) Samples of lossy measurements. Each image is a blurred noisy version of the original. Samples produced by (middle) a baseline that uses Wiener deconvolution, and (right) our model.

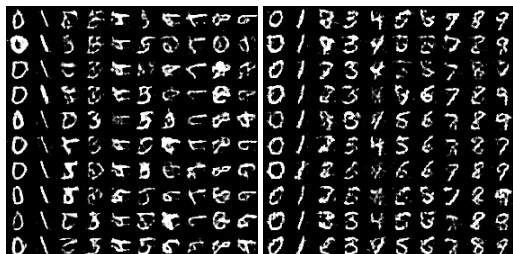

(b) Samples produced by our model trained from two 1D projections of each image. On left, the training data does not include the angle of the projections, so it cannot identify orientation or chirality. On right, the training data includes the angle.

Figure 3: Results with Convolve+Noise on celebA (left) and 1D-projections on MNIST (right).

Learning a GAN on images denoised by Wiener deconvolution leads to poor sample quality while our models are able to produce cleaner samples.

In Fig. 3b, we consider learning a generative model on the 2D images in the MNIST handwritten digit dataset [LeCun et al. (1998)] from pairs of 1D projections. That is, measurements consist of picking two random lines and projecting the image onto each line, so the observed value along the line is the sum of all pixels that project to that point. We consider two variants: in the first, the choice of line is forgotten, while in the second the measurement includes the choice of line. We find for both variants that AmbientGAN recovers a lot of the underlying structure, although the first variant cannot identify the distribution up to rotation or reflection.

## 2 RELATED WORK

There are two distinct approaches to constructing neural network based implicit generative models; autoregressive [Kingma & Welling (2013); Oord et al. (2016b;a)], and adversarial [Goodfellow et al. (2014)]. Some combination approaches have also been successful [Mescheder et al. (2017)].

The adversarial framework has been shown to be extremely powerful in modeling complex data distributions such as images [Radford et al. (2015); Arjovsky et al. (2017); Berthelot et al. (2017)], video [Liang et al. (2017); Vondrick et al. (2016)], and 3D models [Achlioptas et al. (2017); Wu et al. (2016)]. A learned generative model can be useful for many applications. A string of papers [Bora et al. (2017); Zhu et al. (2016); Yeh et al. (2016)] explore the utility of generative priors to solve ill-posed inverse problems. [Shrivastava et al. (2016)] demonstrate that synthetic data can be made more realistic using GANs. [Isola et al. (2016)] and [Zhu et al. (2017)] show how to translate images from one domain to another using GANs.

The idea of operating generators and discriminators on different spaces has been proposed before. [Neyshabur et al. (2017)] explores an interesting connection of training stability with low dimensional projections of samples. They show that training a generator against an array of discriminators, each operating on a different low-dimensional projection of the data can improve stability. Our work is also closely related to [Gadelha et al. (2016)] where the authors create 3D object shapes from a dataset of 2D projections. We note that their setup is a special case of the AmbientGAN framework where the measurement process creates 2D projections using weighted sums of voxel occupancies.

## 3 NOTATION AND OUR APPROACH

Throughout, we use superscript '$r$' to denote real or true distribution, superscript '$g$' for the generated distributions, '$x$' for the underlying space and '$y$' for measurements. Let $p_x^r$ be a real underlying distribution over $\mathbb{R}^n$. We observe lossy measurements performed on samples from $p_x^r$. If we let $m$ be the size of each observed measurement, then, each measurement is an output of some measurement function $f_\theta : \mathbb{R}^n \to \mathbb{R}^m$, parameterized by $\theta$. We allow the measurement function to be stochastic by letting the parameters of the measurement functions have a distribution $p_\theta$. With this notation, for a given $x$ and $\theta$, the measurements are given by $y = f_\theta(x)$. We assume that it is easy to sample $\Theta \sim p_\theta$ and to compute $f_\theta(x)$ for any $x$ and $\theta$. The distributions $p_x^r$ and $p_\theta$ naturally induce a distribution over the measurements $y$ which we shall denote by $p_y^r$. In other words, if $X \sim p_x^r$ and $\Theta \sim p_\theta$, then $Y = f_\Theta(X) \sim p_y^r$.

Our task is the following: there is some unknown distribution $p_x^r$ and a known distribution $p_\theta$. We are given a set of IID realizations $\{y_1, y_2, \ldots, y_s\}$ from the distribution $p_y^r$. Using these, our goal is to create an implicit generative model of $p_x^r$, *i.e.*, a stochastic procedure that can sample from $p_x^r$.

Our main idea is to combine the measurement process with the adversarial training framework, as shown in Fig. 1. Just like in the standard GAN setting, let $Z \in \mathbb{R}^k$, $Z \sim p_z$ be a random latent vector for a distribution $p_z$ that is easy to sample from, such as IID Gaussian or IID uniform. Let $G : \mathbb{R}^k \to \mathbb{R}^n$ be a generator. Let $X^g = G(Z)$, and let $p_x^g$ be the distribution of $X^g$. Thus, our goal is to learn a generator $G$ such that $p_x^g$ is close to $p_x^r$.

However, unlike the standard GAN setting, we do not have access to the desired objects ($X \sim p_x^r$). Instead, we only have a dataset of measurements (samples from $Y \sim p_y^r$). Our main idea is to simulate random measurements on the generated objects $X^g$, and use the discriminator to distinguish real measurements from fake measurements. Thus, we sample a random measurement function $f_\Theta$ by sampling $\Theta \sim p_\theta$ and apply it on $X^g$ to obtain $Y^g = f_\Theta(X^g) = f_\Theta(G(Z))$. Let $p_y^g$ be the distribution of $Y^g$. We set up the discriminator to predict if a given $y$ is a sample from the real measurement distribution $p_y^r$ as opposed to the generated measurement distribution $p_y^g$. Thus, the discriminator is a function $D : \mathbb{R}^m \to \mathbb{R}$.

We let $q(\cdot)$ be the quality function that is used to define the objective, based on the discriminator output. For vanilla GAN, $q(x) = \log(x)$ and for Wasserstein GAN [Arjovsky et al. (2017)], $q(x) = x$. Accordingly, the AmbientGAN objective is the following:

$$\min_G \max_D \mathbb{E}_{Y^r \sim p_y^r}[q(D(Y^r))] + \mathbb{E}_{Z \sim p_z, \Theta \sim p_\theta}[q(1 - D(f_\Theta(G(Z))))].$$

We additionally require $f_\theta$ to be differentiable with respect to its inputs for all $\theta$. We implement $G$ and $D$ as feedforward neural networks. With these assumptions, our model is end-to-end differentiable and can be trained using an approach similar to the standard gradient-based GAN training procedure. In each iteration, we sample $Z \sim p_z$, $\Theta \sim p_\theta$, and $Y^r \sim \text{UNIF}\{y_1, y_2, \ldots, y_s\}$ to use them to compute stochastic gradients of the objective with respect to parameters in $G$ and $D$ by backpropagation. We alternate between updates to parameters of $D$ and updates to parameters of $G$.

We note that our approach is compatible with and complementary to the various improvements proposed to the GAN objective, network architectures, and the training procedures. Additionally, we can easily incorporate additional information, such as per sample labels, in our framework through conditional versions of the generator and discriminator. This is exemplified in our experiments, where we use unconditional and conditional versions of DCGAN [Radford et al. (2015)], unconditional Wasserstein GAN with gradient penalty [Gulrajani et al. (2017)], and an Auxiliary Classifier Wasserstein GAN [Odena et al. (2016)] with gradient penalty.

## 4 MEASUREMENT MODELS

Now, we describe the measurement models that we use for our theoretical and empirical results. We primarily focus on 2D images and thus our measurement models are tailored to this setting. The AmbientGAN learning framework, however, is more general and can be used for other data formats and other measurement models as well. For the rest of this section, we assume that input to the measurement function $(x)$ is a 2D image. We consider the following measurement models:

**Block-Pixels**: Each pixel is independently set to zero with probability $p$. **Convolve+Noise**: Let $k$ be a convolution kernel and let $\Theta \sim p_\theta$ be the distribution of noise. Then the measurements are given by $f_\Theta(x) = k * x + \Theta$, where $*$ is the convolution operator. **Block-Patch**: A randomly chosen $k \times k$ patch is set to zero. **Keep-Patch**: All pixels outside a randomly chosen $k \times k$ patch are set to zero. **Extract-Patch**: A random $k \times k$ patch is extracted. Note that unlike the previous measurement function, the information about the location of the patch is lost. **Pad-Rotate-Project**: We pad the image on all four sides by zeros. Then we rotate the image by a random angle $(\theta)$ about its center. The padding is done to make sure that the original pixels stay within the boundary. Finally, for each channel in the image, we sum the pixels along the vertical axis to get one measurement vector. **Pad-Rotate-Project-$\theta$**: This is the same as the previous measurement function, except that along with the projection values, the chosen angle is also included in the measurements. **Gaussian-Projection**: We project onto a random Gaussian vector which is included in the measurements. So, $\Theta \sim \mathcal{N}(0, I_n)$, and $f_\Theta(x) = (\Theta, \langle \Theta, x \rangle)$.

## 5 THEORETICAL RESULTS

We show that we can provably recover the true underlying distribution $p_x^r$ for certain measurement models. Our broad approach is to show that there is a *unique* distribution $p_x^r$ consistent with the observed measurement distribution $p_y^r$, *i.e.*, the mapping of distributions of samples $p_x^r$ to distribution of measurements $p_y^r$ is invertible even though the map from an individual image $x$ to its measurements $f_\theta(x)$ is not. If this holds, then the following lemma immediately gives a consistency guarantee with the AmbientGAN training procedure.

**Lemma 5.1.** As in Section 3, let $p_x^r$ be the data distribution, $p_\theta$ be the distribution over parameters of the measurement functions and $p_y^r$ be the induced measurement distribution. Further, assume that for the given $p_\theta$, there is a unique probability distribution $p_x^r$ that induces the given measurement distribution $p_y^r$. Then, for the vanilla GAN model [Goodfellow et al. (2014)], if the Discriminator $D$ is optimal, so that $D(\cdot) = \frac{p_y^r(\cdot)}{p_y^r(\cdot) + p_y^g(\cdot)}$, then a generator $G$ is optimal iff $p_x^g = p_x^r$.

All proofs including this one are deferred to Appendix A. Note that the previous lemma makes a non-trivial assumption of uniqueness of the true underlying distribution given the measurement distribution. The next few theorems show that this assumption is satisfied under Gaussian-Projection, Convolve+Noise and Block-Pixels measurement models, thus showing that that we can recover the true underlying distribution with the AmbientGAN framework.

**Theorem 5.2.** For the Gaussian-Projection measurement model (Section 4), there is a unique underlying distribution $p_x^r$ that can induce the observed measurement distribution $p_y^r$.

**Theorem 5.3.** Let $\mathcal{F}(\cdot)$ denote the Fourier transform and let $\text{supp}(\cdot)$ be the support of a function. Consider the Convolve+Noise measurement model (Section 4) with the convolution kernel $k$ and additive noise distribution $p_\theta$. If $\text{supp}(\mathcal{F}(k))^c = \phi$ and $\text{supp}(\mathcal{F}(p_\theta))^c = \phi$, then there is a unique distribution $p_x^r$ that can induce the measurement distribution $p_y^r$.

We remark that the required conditions in the preceding theorem are easily satisfied for the common setting of Gaussian blurring kernel with additive Gaussian noise. The same guarantee can be generalized for any continuous and invertible function instead of a convolution. We omit the details.

Our next theorem makes an assumption of a finite discrete set of pixel values. This assumption holds in most practical scenarios since images are represented with a finite number of discrete values per

channel. In this setting, in addition to a consistency guarantee, we also give a sample complexity result for approximately learning the distributions in the AmbientGAN framework.

**Theorem 5.4.** Assume that each image pixel takes values in a finite set P. Thus $x \in P^n \subset \mathbb{R}^n$. Assume $0 \in P$, and consider the Block-Pixels measurement model (Section 4) with $p$ being the probability of blocking a pixel. If $p < 1$, then there is a unique distribution $p_x^r$ that can induce the measurement distribution $p_y^r$. Further, for any $\epsilon > 0$, $\delta \in (0, 1]$, given a dataset of

$$s = \Omega \left( \frac{|P|^{2n}}{(1-p)^{2n}\epsilon^2} \log \left( \frac{|P|^n}{\delta} \right) \right)$$

IID measurement samples from $p_y^r$, if the discriminator $D$ is optimal, then with probability $\geq 1 - \delta$ over the dataset, any optimal generator $G$ must satisfy $d_{TV}(p_x^g, p_x^r) \leq \epsilon$, where $d_{TV}(\cdot, \cdot)$ is the total variation distance.

## 6 DATASETS AND MODEL ARCHITECTURES

We used three datasets for our experiments. MNIST is a dataset of $28 \times 28$ images of handwritten digits [LeCun et al. (1998)]. CelebA is a dataset of face images of celebrities [Liu et al. (2015)]. We use an aligned and cropped version where each image is $64 \times 64$ RGB. The CIFAR-10 dataset consists of $32 \times 32$ RGB images from 10 different classes [Krizhevsky & Hinton (2009)].

We briefly describe the generative models we used for our experiments. More details on architectures and hyperparameters can be found in the appendix. For the MNIST dataset, we use two GAN models. The first model is a conditional DCGAN which follows the architecture in [Radford et al. (2015)][1], while the second model is an unconditional Wasserstein GAN with gradient penalty (WGANGP) which follows the architecture in [Gulrajani et al. (2017)][2]. For the celebA dataset, we use an unconditional DCGAN and follow the architecture in [Radford et al. (2015)][3]. For the CIFAR-10 dataset, we use an Auxiliary Classifier Wasserstein GAN with gradient penalty (ACWGANGP) which follows the residual architecture in [Gulrajani et al. (2017)][4].

For measurements with 2D outputs, *i.e.* Block-Pixels, Block-Patch, Keep-Patch, Extract-Patch, and Convolve+Noise (see Section 4), we use the same discriminator architectures as in the original work. For 1D projections, *i.e.* Pad-Rotate-Project, Pad-Rotate-Project-$\theta$, we use fully connected discriminators. The architecture of the fully connected discriminator used for the MNIST dataset was 25-25-1 and for the celebA dataset was 100-100-1.

## 7 BASELINES

Now, we describe some baseline approaches that we implemented to evaluate the relative performance of the AmbientGAN framework. Recall that we have a dataset of IID samples $\{y_1, y_2, \ldots y_s\}$ from the measurement distribution $p_y^r$ and our goal is to create an implicit generative model for $p_x^r$.

A crude baseline is to ignore that any measurement happened at all. In other words, for cases where the measurements lie in the same space as the full-samples (for example Convolve+Noise) we can learn a generative model directly on the measurements and test how well it approximates the true distribution $p_x^r$. We call this the "ignore" baseline.

A stronger baseline is based on the following observation: If the measurement functions $f_\theta$ were invertible, and we observed $\theta_i$ for each measurement $y_i$ in our dataset, we could just invert the functions to obtain full-samples $x_i = f_{\theta_i}^{-1}(y_i)$. Then we could directly learn a generative model using these full-samples. Notice that both assumptions are violated in the AmbientGAN setting. First, we may not observe $\theta_i$ and second, the functions may not be invertible. Indeed all the measurement models in Section 4 violate one of the assumptions. However, we can try to *approximate* an inverse function and use the inverted samples to train a generative model. Thus, given a measurement $y_i = f_{\theta_i}(x_i)$, we try to "unmeasure" it and obtain $\hat{x}_i$, an estimate of $x_i$. We then learn a generative model with the estimated inverse samples and test how well it approximates $p_x^r$.

---

[1,3]code reused from `https://github.com/carpedm20/DCGAN-tensorflow`
[2,4]code reused from `https://github.com/igul222/improved_wgan_training`

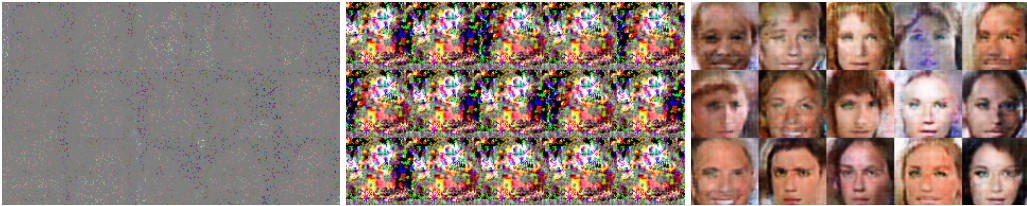

Figure 4: Results with Block-Pixels on celebA. (left) Samples of lossy measurements. Each pixel is blocked independently with probability $p = 0.95$. Samples produced by (middle) unmeasure-blur baseline, and (right) our model.

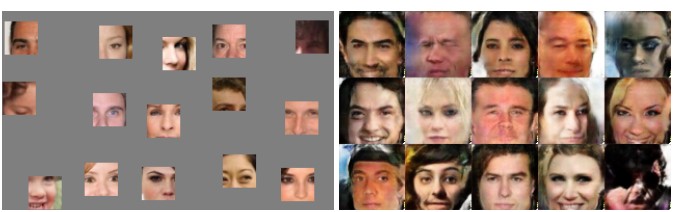

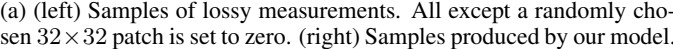

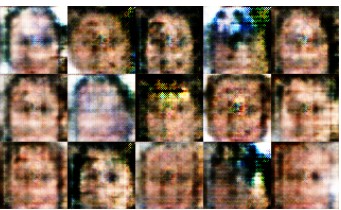

(a) (left) Samples of lossy measurements. All except a randomly chosen $32 \times 32$ patch is set to zero. (right) Samples produced by our model.

(b) Samples produced by our model with Pad-Rotate-Project-$\theta$ measurements.

Figure 5: Results on celebA with (a) Keep-Patch, and (b) 1D projections.

For the measurement models described in Section 4, we now describe the methods we used to obtain approximate inverse functions: (a) For the Block-Pixels measurements, a simple approximate inverse function is to just blur the image so that zero pixels are filled in from the surrounding. We also implemented a more sophisticated approach to fill in the pixels by using total variation inpainting. (b) For Convolve+Noise measurements with a Gaussian kernel and additive Gaussian Noise, we approximate the inverse by a Wiener deconvolution. (c) For Block-Patch measurements, we use the Navier Stokes based inpainting method [Bertalmio et al. (2001)] to fill in the zero pixels.

For other measurement models, it is unclear how to obtain an approximate inverse function. For the Keep-Patch measurement model, no pixels outside a box are known and thus inpainting methods are not suitable. Inverting Extract-Patch measurements is even harder since the information about the position of the patch is also lost. For the Pad-Rotate-Project-$\theta$ measurements, a conventional technique is to sample many angles, and use techniques for inverting the Radon transform [Deans (2007)]. However, since we observe only a few projections at a time, these methods aren't readily applicable. Inverting Pad-Rotate-Project measurements is even harder since it lacks information about $\theta$. So, on this subset of experiments, we report only the results with the AmbientGAN models.

## 8 QUALITATIVE RESULTS

We present some samples generated by the baselines and our models. For each experiment, we show the samples from the dataset of measurements ($Y^r$) available for training, samples generated by the baselines (when applicable) and the samples generated by our models ($X^g$). We show samples only for a selected value of parameter settings. More results are provided in the appendix. All results on MNIST are deferred to the appendix.

**Block-Pixels**: Fig. 4 shows results on celebA with DCGAN and Fig. 6 on CIFAR-10 with ACW-GANGP. We see that the samples are heavily degraded in our measurement process (left image). Thus, it is challenging for baselines to invert the measurements process, and correspondingly, they do not produce good samples (middle image). Our models are able to produce images with good visual quality (right image).

**Convolve+Noise**: We use a Gaussian kernel and IID Gaussian noise. Fig. 3a shows results on celebA with DCGAN. We see that the measurements are drowned in noise (left image) and the baselines

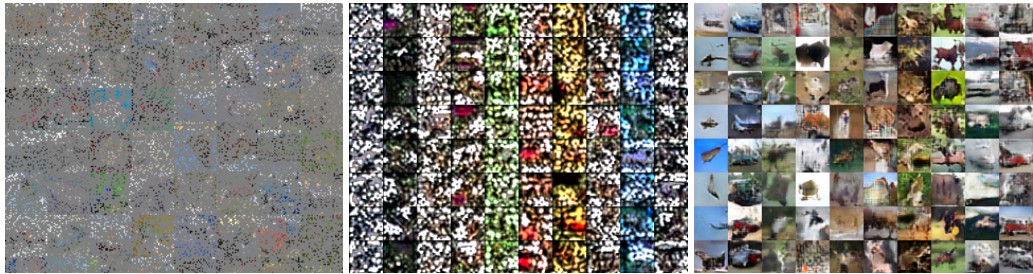

Figure 6: Results with Block-Pixels on CIFAR-10. (left) Samples of lossy measurements. Each pixel is blocked independently with probability $p = 0.8$. Samples produced by (middle) unmeasure-blur baseline, and (right) our model.

struggle to extract the original image, giving samples of low quality (middle image). Our models are able to produce samples with clear faces (right image).

**Block-Patch, Keep-Patch**: Fig. 2 shows the results for Block-Patch and Fig. 5a for Keep-Patch measurements on celebA with DCGAN. On both measurement distributions, our models are able to create coherent faces (right image) by observing only parts of one image at a time.

**1D projections**: Pad-Rotate-Project and Pad-Rotate-Project-$\theta$ measurement models exhibit drastic signal degradation; most of the information in a sample is lost during the measurements process. For our experiments, we use *two* measurements at a time. Fig. 3b shows the results on MNIST with DCGAN. While the first model is able to learn only up to rotation and reflection (left image), we note that generated digits have similar orientations and chirality within each class without any explicit incentive. We hypothesize that the model prefers this mode because it is easier to learn with consistent orientation per class. The second measurement model contains the rotation angle and thus produces upright digits (right image). While in both cases, the generated images are of lesser visual quality, our method demonstrates that we can produce images of digits given only 1D projections.

**Failure case**: In Fig. 5b, we show the samples obtained from our model trained on celebA dataset with Pad-Rotate-Project-$\theta$ measurements with a DCGAN. We see that the model has learned a very crude outline of a face, but lacks details. This highlights the difficulty in learning complex distributions with just 1D projections and a need for better understanding of distribution recovery under projection measurement model as well as better methods for training GANs.

## 9    QUANTITATIVE RESULTS

We report inception scores [Salimans et al. (2016)] to quantify the quality of the generative models learned in the AmbientGAN framework. For the CIFAR-10 dataset, we use the Inception model[Szegedy et al. (2016)] trained on the ImageNet dataset [Deng et al. (2009)][1]. For computing a similar score on MNIST, we trained a classification model with two conv+pool layers followed by two fully connected layers[2]. The final test set accuracy of this model was $99.2\%$.

### 9.1    MNIST

For Block-Pixels measurements on MNIST, we trained several models with our approach and the baselines, each with a different probability $p$ of blocking pixels. For each model, after convergence, we computed the inception score using the network described above. A plot of the inception scores as a function of $p$ is shown in Fig. 7 (left). We note that at $p = 0$, *i.e.* if no pixels are blocked, our model is equivalent to a conventional GAN. As we increase $p$, the baseline models quickly start to perform poorly, while the AmbientGAN models continue to perform relatively well.

---

[1]model weights reused from `http://download.tensorflow.org/models/image/imagenet/inception-2015-12-05.tgz`

[2]code reused from `https://www.tensorflow.org/get_started/mnist/pros`

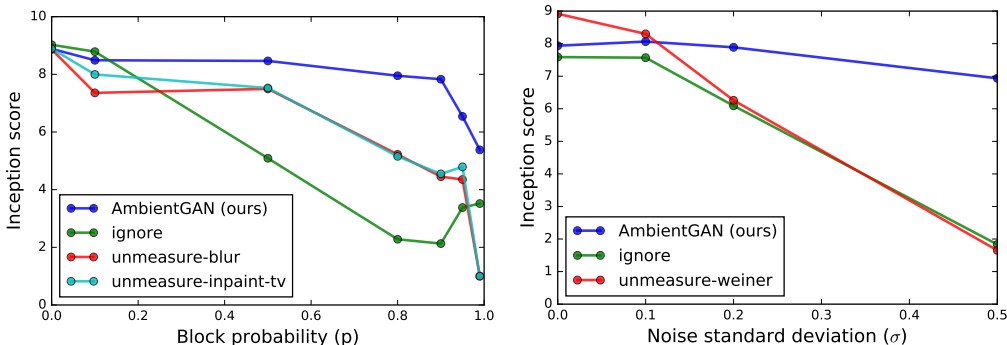

Figure 7: Results on MNIST with WGANGP. (left) Block-Pixels (right) Convolve+Noise

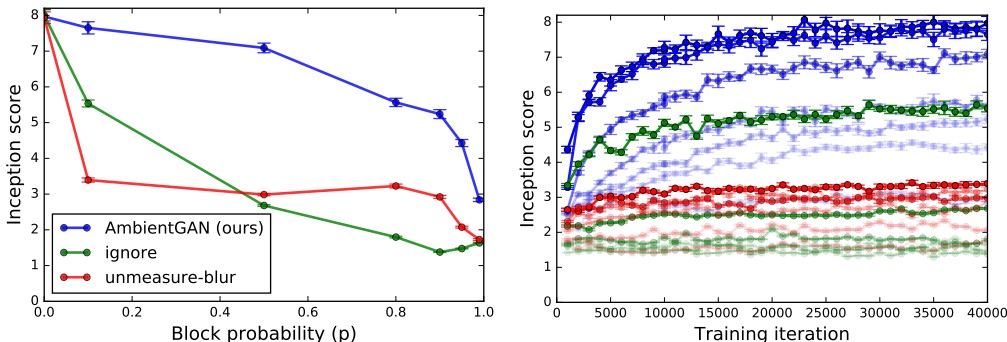

Figure 8: Quantitative results on CIFAR-10 with ACWGANGP, Block-Pixels measurement. (left) Inception score vs blocking probability $p$. (right) Inception score vs training iteration with darkness proportional to $1 - p$. Vertical bars indicate $95\%$ confidence intervals.

For the Convolve+Noise measurements with a Gaussian kernel of radius 1 pixel, and additive Gaussian noise with zero mean and standard deviation $\sigma$, we trained several models on MNIST by varying the value of $\sigma$. A plot of the inception score as a function of $\sigma$ is shown in Fig. 7 (right). We see that for small variance of additive noise, Wiener deconvolution and the "ignore" baseline perform quite well. However, as we start to increase the noise levels, these baselines quickly deteriorate in performance, while the AmbientGAN models maintain a high inception score.

For 1D projection measurements, we report the inception scores for the samples produced by the AmbientGAN models trained with two projection measurements at a time. The Pad-Rotate-Project model produces digits at various orientations and thus does quite poorly, achieving an inception score of just $4.18$. The model with Pad-Rotate-Project-$\theta$ measurements produces well-aligned digits and achieves an inception score of $8.12$. For comparison, the vanilla GAN model trained with fully-observed samples achieves an inception score of $8.99$. Thus, the second model comes quite close to the performance of the fully-observed case while being trained only on 1D projections.

## 9.2 CIFAR-10

In Fig. 8 (left), we show a plot of inception score vs the probability of blocking pixels $p$ in the Block-Pixels measurement model on CIFAR-10. We note that the total variation inpainting method is quite slow and the performance on MNIST was about the same as unmeasure-blur baseline. So, we do not run inpainting baselines on the CIFAR-10 dataset. From the plots, we see a trend similar to the plot obtained with MNIST (Fig. 7, left), showing the superiority of our approach over baselines. We show the inception score as a function of training iteration in Fig. 8 (right).

## 10   CONCLUSION

Generative models are powerful tools, but constructing a generative model requires a large, high-quality dataset of the distribution of interest. We show how to relax this requirement, by learning a distribution from a dataset that only contains incomplete, noisy measurements of the distribution. We hope that this will allow for the construction of new generative models of distributions for which no high-quality dataset exists.

## ACKNOWLEDGEMENTS

We would like to thank Philipp Krähenbühl, Ajil Jalal, Surbhi Goel, and Jessica Hoffmann for helpful discussions. This research has been supported by NSF Grants CCF 1407278, 1422549, 1618689, DMS 1723052, ARO YIP W911NF-14-1-0258 and NVIDIA Corporation.

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

## APPENDIX A

### 10.1 PROOF OF LEMMA 5.1

**Lemma.** As in Section 3, let $p_x^r$ be the data distribution, $p_\theta$ be the distribution over parameters of the measurement functions and $p_y^r$ be the induced measurement distribution. Further, assume that for the given $p_\theta$, there is a unique probability distribution $p_x^r$ that induces the given measurement distribution $p_y^r$. Then, for the vanilla GAN model [Goodfellow et al. (2014)], if the Discriminator $D$ is optimal, so that

$$D(\cdot) = \frac{p_y^r(\cdot)}{p_y^r(\cdot) + p_y^g(\cdot)},$$

then a generator $G$ is optimal iff $p_x^g = p_x^r$.

*Proof.* From the same argument as in Theorem 1 in [Goodfellow et al. (2014)], it follows that $p_y^g = p_y^r$. Then, since there is a unique probability distribution $p_x^r$ that can induce the given measurement distribution, it follows that $p_x^g = p_x^r$. The converse is trivially true. □

### 10.2 PROOF OF THEOREM 5.2

**Theorem.** For the Gaussian-Projection measurement model (Section 4), there is a unique underlying distribution $p_x^r$ that can induce the observed measurement distribution $p_y^r$.

*Proof.* We note that Since $\Theta \sim \mathcal{N}(0, I_n)$, all possible directions for projections are covered. Further, since the measurement model includes the projection vector $\Theta$ as a part of the measurements, in order to match the measurement distribution, the underlying distribution $p_x^r$ must be such that all 1D marginals are matched. Thus, by Cramer-Wold theorem [Cramér & Wold (1936)], any sequence of random vectors that match the 1D marginals must converge in distribution to the true underlying distribution. Thus, in particular, there is a unique probability distribution $p_x^r$ that can match all 1D marginals obtained with the Gaussian projection measurements. □

### 10.3 PROOF OF THEOREM 5.3

**Theorem.** Let $\mathcal{F}(\cdot)$ denote the Fourier transform and let $\mathrm{supp}(\cdot)$ be the support of a function. Consider the Convolve+Noise measurement model (Section 4) with the convolution kernel $k$ and additive noise distribution $p_\theta$. If $\mathrm{supp}(\mathcal{F}(k))^c = \phi$ and $\mathrm{supp}(\mathcal{F}(p_\theta))^c = \phi$, then there is a unique distribution $p_x^r$ that can induce the measurement distribution $p_y^r$.

*Proof.* Let $X \sim p_x$. Let $\Theta \sim p_\theta$. Let $Z = k * X$ so that $Z \sim p_z$, and $Y = Z + \Theta$, so $Y \sim p_y$. With a slight abuse of notation, we will denote the probability density functions (pdf) also by $p$ subscripted with the variable name. Then we have

$$Z = k * X,$$
$$\Leftrightarrow \mathcal{F}(Z) = \mathcal{F}(k)\mathcal{F}(X),$$
$$\Leftrightarrow \mathcal{F}(X) = \mathcal{F}(Z)/\mathcal{F}(k),$$
$$\Leftrightarrow X = \mathcal{F}^{-1}(\mathcal{F}(Z)/\mathcal{F}(k)),$$

where the penultimate step follows since by assumption, $\mathcal{F}(k)$ is nowhere 0. In the last step, $\mathcal{F}^{-1}$ is the inverse Fourier transform.

Thus, there is a bijective map between $X$ and $Z$. Since the Fourier and the inverse Fourier are continuous transformations, this map is also continuous. So, we can write $Z = h(X)$, where $h$ is a bijective, differentiable function. So, the pdfs of $X$ and $Z$ are related as

$$p_x(\cdot) = p_z(h(\cdot)) \left|\det(J_h(\cdot))\right|,$$

where $J_h(\tilde{x})$ is the Jacobian of $h$ evaluated at $\tilde{x}$.

Now, note that since Y is a sum of two random variables, its pdf is a convolution of the individual probability density functions. So we have:

$$p_y = p_z * p_\theta$$

Taking the Fourier transform on both sides, we have

$$\mathcal{F}(p_y) = \mathcal{F}(p_z)\mathcal{F}(p_\theta),$$
$$\Leftrightarrow \mathcal{F}(p_z) = \mathcal{F}(p_y)/\mathcal{F}(p_\theta),$$
$$\Leftrightarrow p_z = \mathcal{F}^{-1}(\mathcal{F}(p_y)/\mathcal{F}(p_\theta)),$$

where the penultimate step follows since by assumption, $\mathcal{F}(p_\theta)$ is nowhere 0.

Combining the two results, we have a reverse map from the measurement distribution $p_y$ to a sample distribution $p_x$. Thus, the reverse map uniquely determines the true underlying distribution $p_x$, concluding the proof. □

## 10.4   PROOF OF THEOREM 5.4

We first state a slightly different version of Theorem 1 from [Goodfellow et al. (2014)] for the discrete setting. We shall use $[n]$ to denote the set $\{1, 2, \ldots n\}$, and use $\mathbb{I}(\cdot)$ to denote the indicator function.

**Lemma 10.1.** Consider a dataset of measurement samples $\{y_1, y_2, \ldots y_s\}$, where each $y_i \in [t]$. We define the empirical version of the vanilla GAN objective as

$$\min_G \max_D \frac{1}{s} \sum_{i=1}^{s} \log(D(y_i)) + \mathbb{E}_{Y^g \sim p_y^g}[\log(1 - D(Y^g))].$$

For $j \in [t]$, let $\hat{p}_y^r(j) = \sum \mathbb{I}(y_i = j)/s$ be the empirical distribution of samples. Then the optimal discriminator for the empirical objective is such that

$$D(\cdot) = \frac{\hat{p}_y^r(\cdot)}{\hat{p}_y^r(\cdot) + p_y^g(\cdot)}.$$

Additionally, if we fix the discriminator to be optimal, then any optimal generator must satisfy $p_y^g = \hat{p}_y^r$.

*Proof.* The Empirical Risk Minimization (ERM) version of the loss is equivalent to the taking expectation of the data dependent term with respect to the empirical distribution. Replacing the real data distribution with the empirical version in the proof of Theorem 1 from [Goodfellow et al. (2014)], we obtain the result. □

Now we give a proof of Theorem 5.4.

**Theorem.** Assume that each image pixel takes values in a finite set P. Thus $x \in P^n \subset \mathbb{R}^n$. Assume $0 \in P$, and consider the Block-Pixels measurement model (Section 4) with $p$ being the probability of blocking a pixel. If $p < 1$, then there is a unique distribution $p_x^r$ that can induce the measurement distribution $p_y^r$. Further, for any $\epsilon > 0$, $\delta \in (0, 1]$, given a dataset of

$$s = \Omega\left(\frac{|P|^{2n}}{(1-p)^{2n}\epsilon^2} \log\left(\frac{|P|^n}{\delta}\right)\right)$$

IID measurement samples from $p_y^r$, if the discriminator $D$ is optimal, then with probability $\geq 1 - \delta$ over the dataset, any optimal generator $G$ must satisfy $d_{TV}(p_x^g, p_x^r) \leq \epsilon$, where $d_{TV}(\cdot, \cdot)$ is the total variation distance.

*Proof.* We first consider a more general case and apply that to the Block-Pixels model. Consider a discrete distribution $p_x$ over $[t]$. We apply random measurement functions to samples from $p_x$ to obtain measurements. Assume that each measurement also belongs to the same set, *i.e.* $[t]$. Let $A \in \mathbb{R}^{t \times t}$ be the transition matrix so that $A_{ij}$ is the probability (under the randomness in measurement functions) that measurement $i$ was produced by sample $j$. Then the distribution over measurements $p_y$ can be written in terms of $p_x$ and $A$ as:

$$p_y = A p_x$$

Thus, if the matrix $A$ is invertible, we can guarantee that the distribution $p_x$ is recoverable from $p_y$.

Assuming $A$ is invertible, we now turn to the sample complexity. Let $\lambda$ be the minimum of magnitude of eigenvalues of $A$. Since $A$ is invertible, $\lambda > 0$. Let the dataset of measurements be $\{y_1, y_2, \ldots y_s\}$. For $j \in [t]$ and for $k \in [s]$, Let $Y_k^j = \mathbb{I}(y_k = j)$. Then for any $\epsilon > 0$, we have

$$
\begin{aligned}
\mathbb{P}\left( \|\hat{p}_y^r - p_y^r\|_2^2 \geq \frac{\lambda^2 \epsilon^2}{t} \right) &= \mathbb{P}\left( \sum_{j=1}^{t} (\hat{p}_y^r(j) - p_y^r(j))^2 \geq \frac{\lambda^2 \epsilon^2}{t} \right), \\
&\leq \mathbb{P}\left( \bigcup_{j=1}^{t} \left\{ (\hat{p}_y^r(j) - p_y^r(j))^2 \geq \frac{\lambda^2 \epsilon^2}{t^2} \right\} \right), \\
&\leq \sum_{j=1}^{t} \mathbb{P}\left( |\hat{p}_y^r(j) - p_y^r(j)| \geq \frac{\lambda \epsilon}{t} \right), \\
&= \sum_{j=1}^{t} \mathbb{P}\left( \left| \frac{\sum_{k=1}^{s} Y_k^j}{s} - p_y^r(j) \right| \geq \frac{\lambda \epsilon}{t} \right), \\
&\leq \sum_{j=1}^{t} 2 \exp(-2s\lambda^2 \epsilon^2 / t^2), \\
&= 2t \exp(-2s\lambda^2 \epsilon^2 / t^2),
\end{aligned}
$$

where we used union bound and Chernoff inequalities. Setting this to $\delta$, we get

$$s = \frac{t^2}{2\lambda^2 \epsilon^2} \log\left( \frac{2t}{\delta} \right).$$

From Lemma 10.1, we know that the optimal generator must satisfy $p_y^g = \hat{p}_y^r$. By invertibility of $A$, we know that $p_x^g = A^{-1} p_y^g$, and that $p_x^r = A^{-1} p_y^r$. Thus, we obtain that with probability $\geq 1 - \delta$,

$$
\begin{aligned}
2 d_{TV}(p_x^g, p_x^r) &= \|p_x^g - p_x^r\|_1, \\
&\leq \sqrt{t} \|p_x^g - p_x^r\|_2, \\
&= \sqrt{t} \|A^{-1} p_y^g - p_x^r\|_2, \\
&= \sqrt{t} \|A^{-1} \hat{p}_y^r - p_x^r\|_2, \\
&= \sqrt{t} \|A^{-1}(p_y^r + \hat{p}_y^r - p_y^r) - p_x^r\|_2, \\
&= \sqrt{t} \|p_x^r + A^{-1}(\hat{p}_y^r - p_y^r) - p_x^r\|_2, \\
&\leq \sqrt{t} \|A^{-1}\|_2 \|\hat{p}_y^r - p_y^r\|_2, \\
&\leq \sqrt{t} \frac{1}{\lambda} \frac{\lambda \epsilon}{\sqrt{t}}, \\
&= \epsilon.
\end{aligned}
$$

Now we turn to the specific case of Block-Pixels measurement. We proceed by dividing the set of all possible $|P|^n$ images into $n+1$ classes. The $i$-th class has those images that have exactly $i$ pixels with zero value. We sort the images according to their class number (arbitrary ordering within the class) and consider the transition matrix $A$. Note that given an image from class $i$ it must have $j \geq i$ zero pixels after the measurement. Also, no image in class $i$ can produce another image in the same class after measurements. Thus, the transition matrix is lower triangular.

Since each pixel is blocked independently with probability $p$ and since there are $n$ pixels, the event that no pixels are blocked occurs with probability $(1-p)^n$. Thus, every image has at least $(1-p)^n$ chance of being unaffected by the measurements. Any unaffected image maps to itself and thus forms diagonal entries in the transition matrix. So, we observe that the diagonal entries of the transition matrix are strictly positive and their minimum value is $(1-p)^n$.

For a triangular matrix, the diagonal entries are precisely the eigenvalues and hence we have proved that $A$ is invertible and the smallest eigenvalue is $(1-p)^n$. Combined with the result above, by setting $\lambda = (1-p)^n$, and $t = |P|^n$, we conclude the proof.

$\square$

## APPENDIX B

### 10.5  MODEL ARCHITECTURE DETAILS

The DCGAN model on MNIST follows the architecture in [Radford et al. (2015)]. The noise input to the generator ($Z$) has 100 dimensions where each coordinate is sampled IID Uniform on $[-1, 1]$. The generator uses two linear layers followed by two deconvolutional layers. The labels are concatenated with the inputs of each layer. The discriminator uses two convolutional layers followed by two linear layers. As with the generator, the labels are concatenated with the inputs of each layer. Batch-norm is used in both generator and the discriminator.

The WGANGP model on MNIST follows the architecture in [Gulrajani et al. (2017)]. The generator takes in a latent vector of 128 dimensions where each coordinate is sampled IID Uniform on $[-1, 1]$. The generator then applies one linear and three deconvolutional layers. The discriminator uses three convolutional layers followed by one linear layer. Batch-norm is not used.

The unconditional DCGAN model on celebA follows the architecture in [Radford et al. (2015)]. The latent vector has 100 dimensions where each coordinate is Uniform on $[-1, 1]$. The generator applies one linear layer followed by four deconvolutional layers. The discriminator uses four convolutional layers followed by a linear layer. Batch-norm is used in both generator and the discriminator.

The ACWGANGP model on CIFAR-10 follows the residual architecture in [Gulrajani et al. (2017)]. The latent vector has 128 dimensions where each coordinate is sampled from IID standard Gaussian distribution. The generator has a linear layer followed by three residual blocks. Each residual block consists of two repetitions of the following three operations: conditional batch normalization followed by a nonlinearity followed by an upconvolution layer. The residual blocks are followed by another conditional batch normalization, a final convolution, and a final tanh non-linearity. The discriminator consists of one residual block with two convolutional layers followed by three residual blocks, and a final linear layer.

## APPENDIX C

Here, we present some more results for various measurement models.

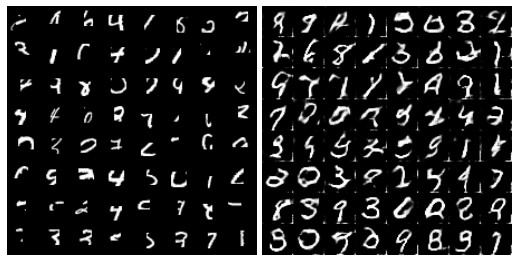 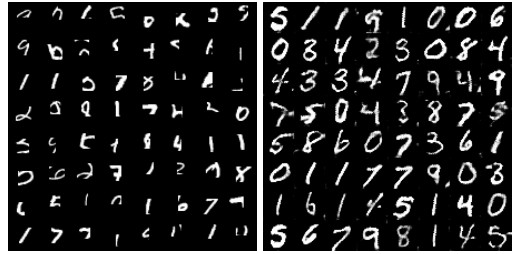

(a) (left) Samples of lossy measurements. All except a randomly chosen $14 \times 14$ patch is set to zero. (right) Samples produced by the our model.

(b) (left) Samples of lossy measurements. A randomly chosen $14 \times 14$ patch is extracted. (right) Samples produced by our model.

Figure 9: Results on MNIST with (a) Keep-Patch, and (b) Extract-Patch

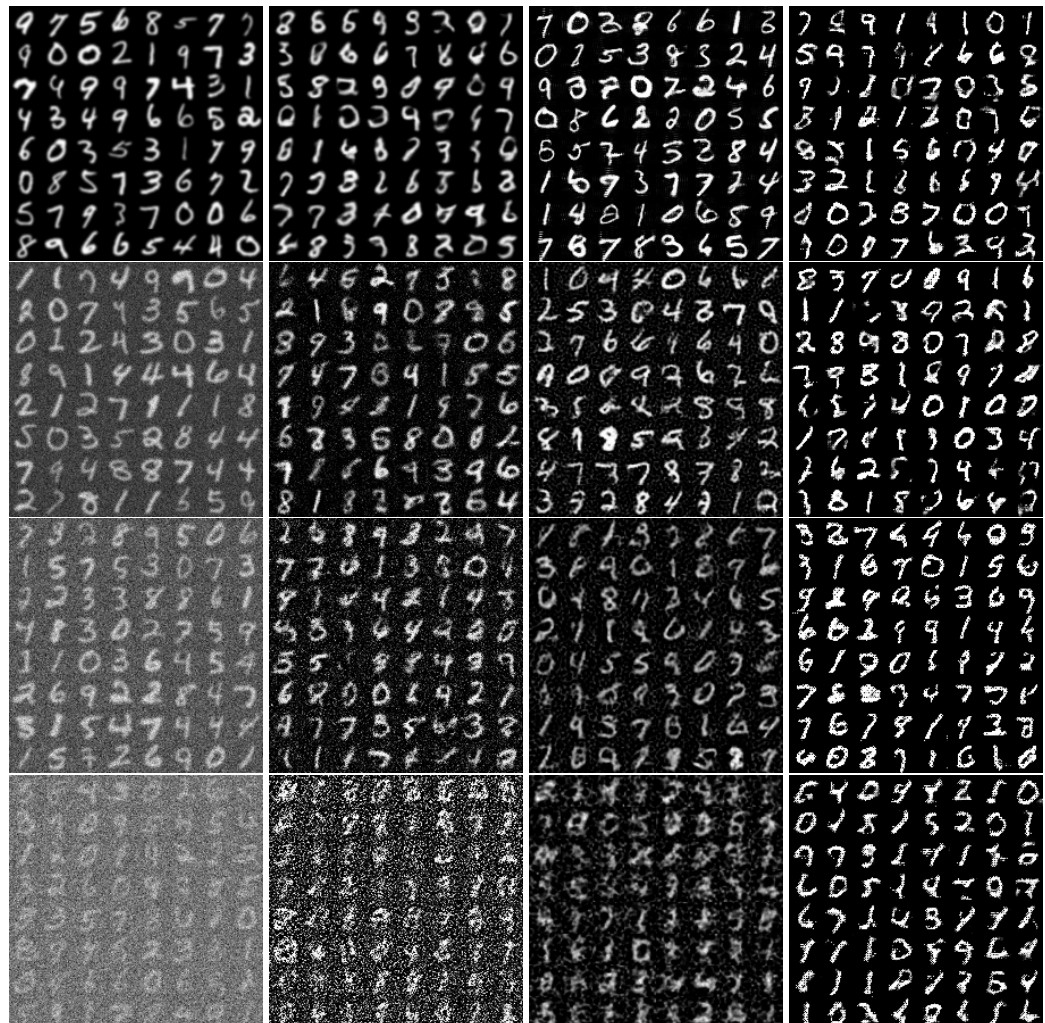

Figure 10: Results with Convolve+Noise on MNIST. Each image is blurred with a Gaussian kernel of radius 1 pixel and noise of std dev $\sigma$ is added. Rows from top to bottom have $\sigma = 0.0$, 0.1, 0.2, and 0.5 respectively. Columns from left to right are: (1) Samples of lossy measurements. (2) Samples produced by ignore baseline. (3) Samples produced by unmeasure-wiener-deconvolution baseline. (4) Samples produced by our model.

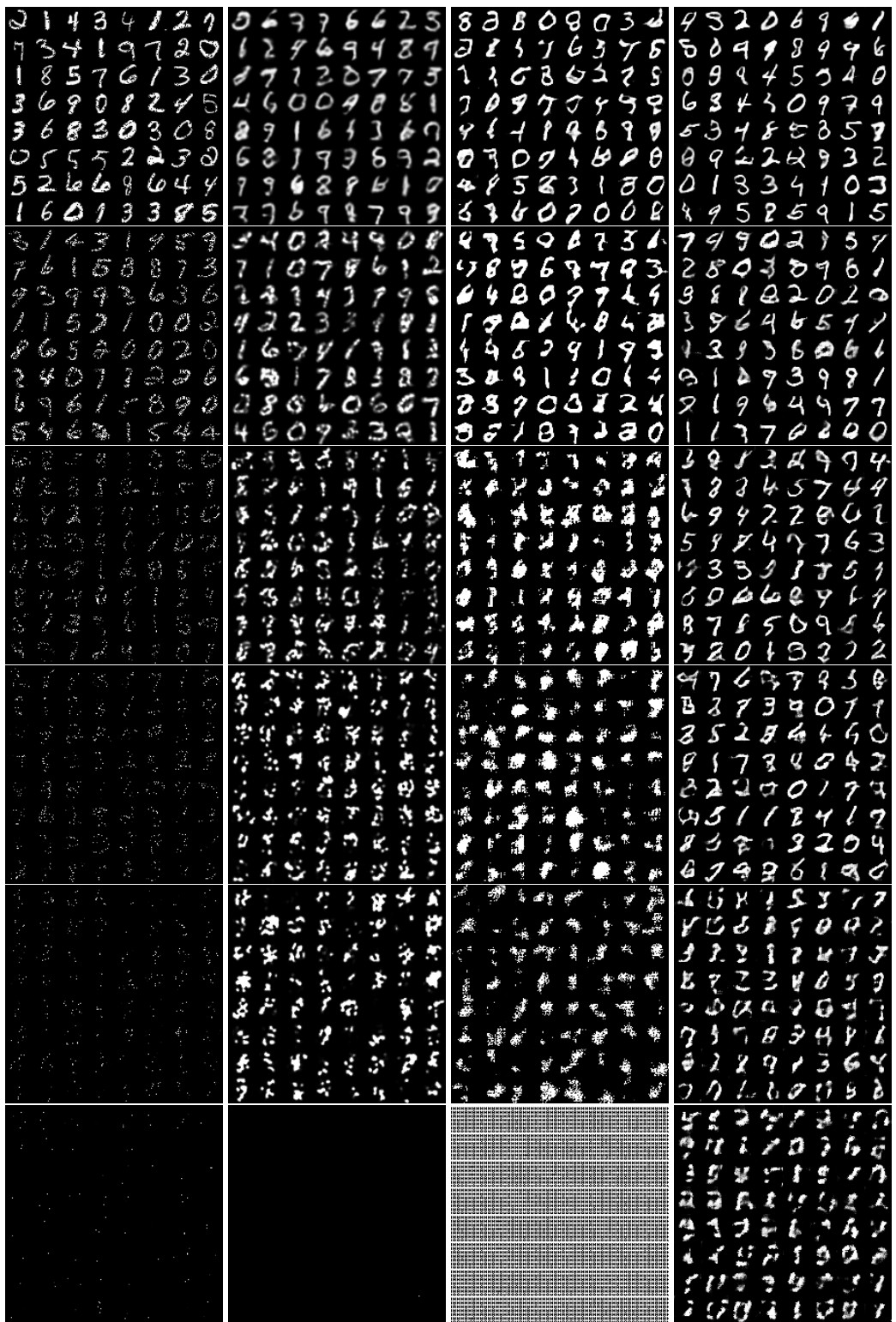

Figure 11: Results with Block-Pixels on MNIST. Rows from top to bottom have blocking probability 0.1, 0.5, 0.8, 0.9, 0.95 and 0.99 respectively. Columns from left to right are: (1) Samples of lossy measurements. (2) Samples produced by unmeasure-blur baseline. (3) Samples produced by unmeasure-inpaint-total-variation baseline. (4) Samples produced by our model.

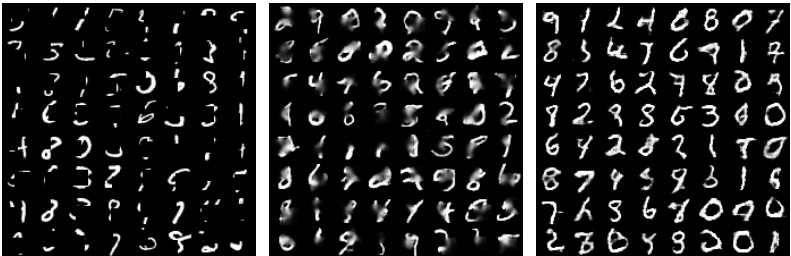

Figure 12: Results with Block-Patch on MNIST. (left) Samples of lossy measurements. A randomly chosen $14 \times 14$ patch is set to zero. (middle) Samples produced by unmeasure-navier-stokes-inpainting baseline. (right) Samples produced by the our model.

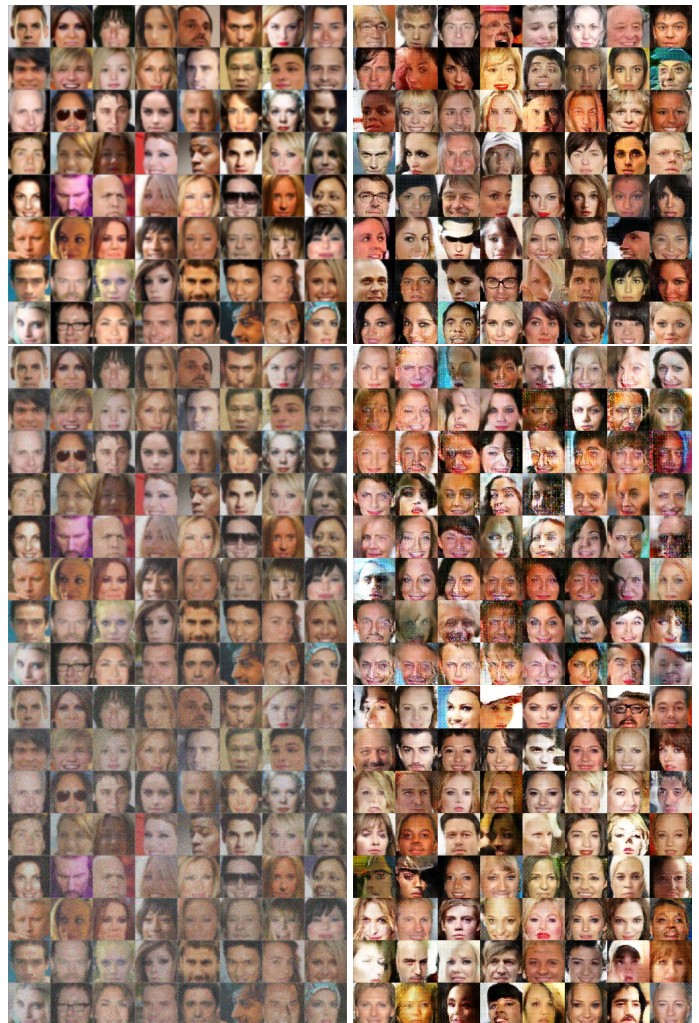

Figure 13: Results with Convolve+Noise on celebA. Each image is blurred with a Gaussian kernel of radius 1 pixel and noise of std dev $\sigma$ is added. Rows from top to bottom have $\sigma = 0.0$, $0.1$, and $0.2$ respectively. The left column shows samples of lossy measurements. The right column shows samples produced by our model.

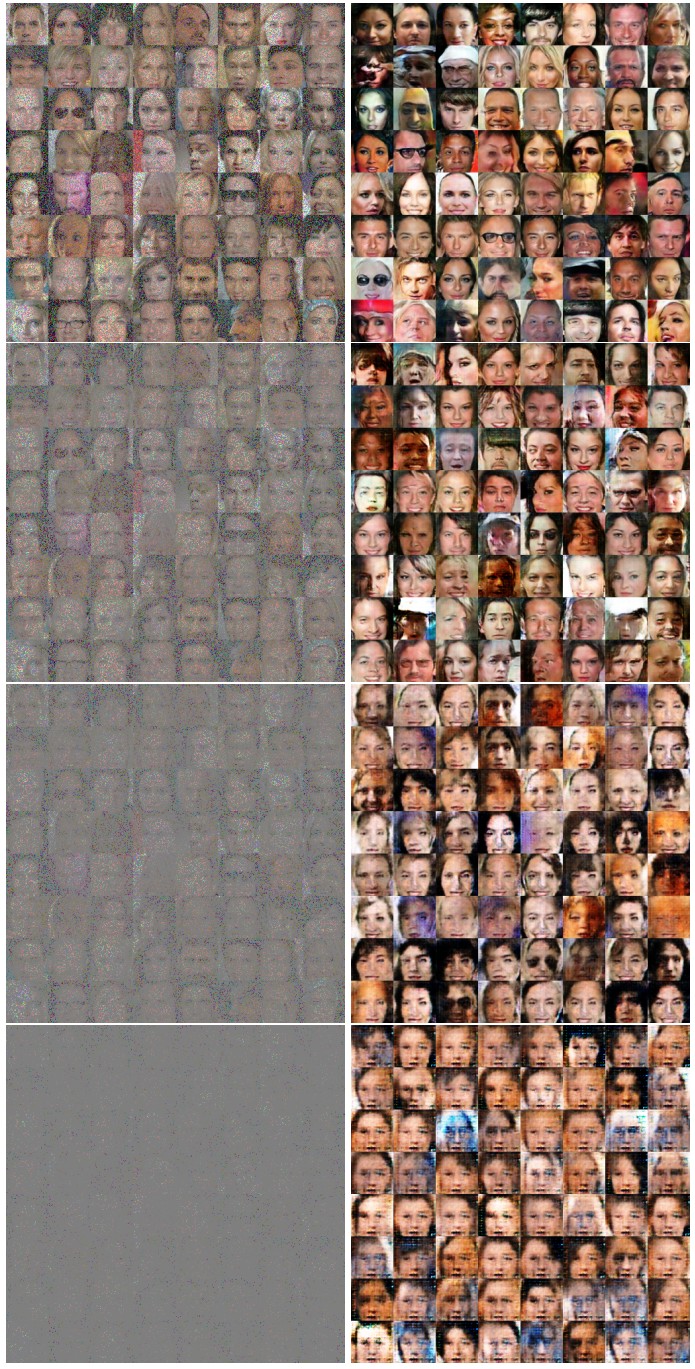

Figure 14: Results with Block-Pixels on celebA. Rows from top to bottom have blocking probability 0.5, 0.8, 0.9, and 0.98 respectively. The left column shows samples of lossy measurements. The right column shows samples produced by our model.

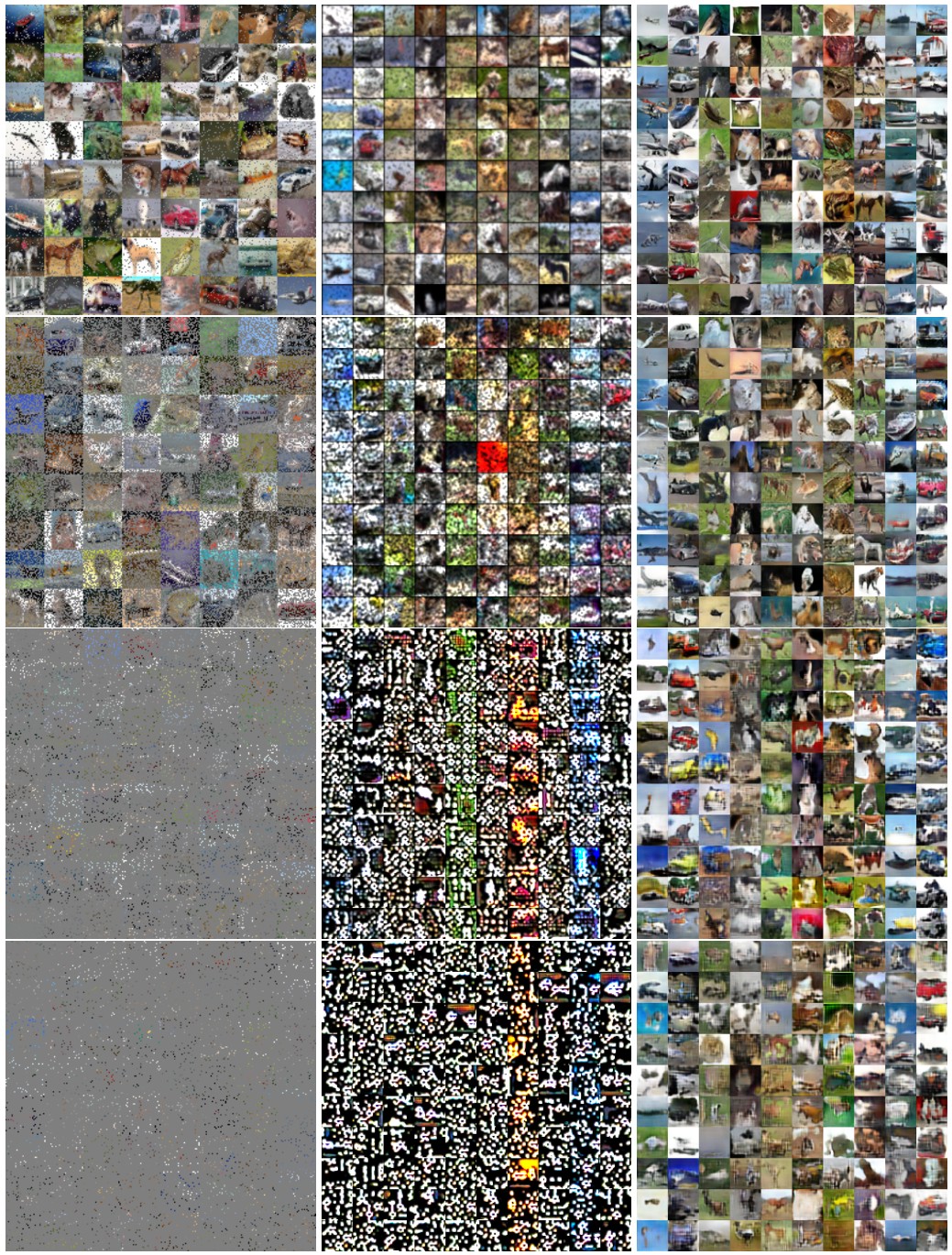

Figure 15: Results with Block-Pixels on CIFAR-10. Rows from top to bottom have blocking probability 0.1, 0.5, 0.9, and 0.95 respectively. Columns from left to right are: (1) Samples of lossy measurements. (2) Samples produced by unmeasure-blur baseline. (3) Samples produced by our model.

APPENDIX D

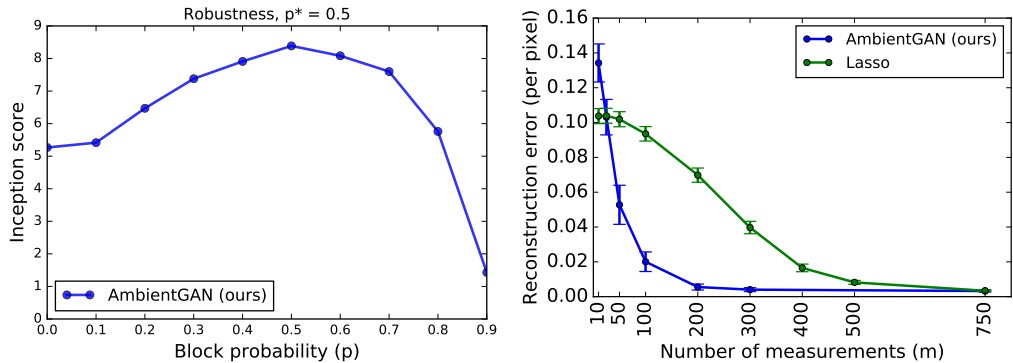

Figure 16: MNIST with WGANGP (left) Robustness experiment with Block-Pixels measurement (right) Compressed sensing using AmbientGAN. Vertical bars indicate 95% confidence intervals.

## 10.6 ROBUSTNESS TO MEASUREMENT MODEL MISMATCH

So far, in our analysis and experiments, we assumed that the parametric form of the measurement function and the distribution of those parameters is exactly known. This was then used for simulating the stochastic measurement process. Here, we consider the case where the parameter distribution is only approximately known. In this case, one would like the training process to be robust, i.e. the quality of the learned generator to be close to the case where the parameter distribution is exactly known. Through the following experiment, we empirically demonstrate that the AmbientGAN approach is robust to systematic mismatches in the parameter distribution of the measurement function.

Consider the Block-Pixels measurement model (Section 4). We use the MNIST dataset. Pixels are blocked with probability $p^* = 0.5$ to obtain a dataset of measurements. For several values of blocking probability $p$ for the measurement function applied to the output of the generator, we train AmbientGAN models with this dataset. After training, we compute the inception score of the learned generators and plot it as a function of $p$ in Fig. 16 (left). We note that the plot peaks at $p = p^* = 0.5$ and gradually drops on both sides. This suggests that our method is somewhat robust to parameter distribution mismatch.

## 10.7 COMPRESSED SENSING USING AMBIENTGAN

We provide further evidence that the generator learned through AmbientGAN approach captures the data distribution well. Generative models have been shown to improve sensing over sparsity-based approaches [Bora et al. (2017)]. We attempt to use the GAN learned using our procedure for compressed sensing.

We trained an AmbientGAN with Block-Pixels measurement model (Section 4) on MNIST with $p = 0.5$. Using the learned generator, we followed the rest of the procedure in [Bora et al. (2017)] using their code[3]. Fig. 16 (right) shows a plot of reconstruction error vs the number of measurements, comparing Lasso with AmbienGAN. Thus, we observe a similar reduction in the number of measurements while using AmbientGAN trained with corrupted samples instead of a regular GAN trained with fully observed samples.

---

[3]reused from https://github.com/AshishBora/csgm

