# OpenReview forum: "AmbientGAN: Generative models from lossy measurements"
_ICLR.cc/2018/Conference — Accept (Oral)_

### Official Review · AnonReviewer2 · 2017-11-23
**A nice paper dealing with an important problem.**

**Rating:** 7
**Confidence:** 4

**Review:**

Quick summary:
This paper shows how to train a GAN in the case where the dataset is corrupted by some measurement noise process. They propose to introduce the noise process into the generation pipeline such that the GAN generates a clean image, corrupts its own output and feeds that into the discriminator. The discriminator then needs to decide whether this is a real corrupted measurement or a generated one.  The method is demonstrated to the generate better results than the baseline on a variety of datasets and noise processes.

Quality:
I found this to be a nice paper - it has an important setting to begin with and the proposed method is clean and elegant albeit a bit simple.

Originality:
I'm pretty sure this is the first paper to tackle this problem directly in general.

Significance:
This is an important research direction as it is not uncommon to get noisy measurements in the real world under different circumstances.

Pros:
* Important problem
* elegant and simple solution
* nice results and decent experiments (but see below)

Cons:
* The assumption that the measurement process *and* parameters are known is quite a strong one. Though it is quite common in the literature to assume this, it would have been interesting to see if there's a way to handle the case where it is unknown (either the process, parameters or both).
* The baseline experiments are a bit limited - it's clear that such baselines would never produce samples which are any better than the "fixed" version which is fed into them. I can't however, think of other baselines other than "ignore" so I guess that is acceptable.
* I wish the authors would show that they get a *useful* model eventually - for example, can this be used to denoise other images from the dataset?

Summary:
This is a nice paper which deals with an important problem, has some nice results and while not groundbreaking, certainly merits a publication.

---

### Official Review · AnonReviewer3 · 2017-11-27
**Paper exploring GAN training under a linear projection measurement model.**

**Rating:** 7
**Confidence:** 4

**Review:**

The paper explores GAN training under a linear measurement model in which one assumes that the underlying state vector $x$ is not directly observed but we do have access to measurements $y$ under a linear measurement model plus noise. The paper explores in detail several practically useful versions of the linear measurement model, such as blurring, linear projection, masking etc. and establishes identifiability conditions/theorems for the underlying models.
The AmbientGAN approach advocated in the paper amounts to learning end-to-end differentiable Generator/Discriminator networks that operate in the measurement space. The experimental results in the paper show that this works much better than reasonable baselines, such as trying to invert the measurement model for each individual training sample, followed by standard GAN training.
The theoretical analysis is satisfactory. However, it would be great if the theoretical results in the paper were able to associate the difficulty of the inversion process with the difficulty of AmbientGAN training. For example, if the condition number for the linear measurement model is high, one would expect that recovering the target real distribution is more difficult. The condition in Theorem 5.4 is a step in this direction, showing that the required number of samples for correct recovery increases with the probability of missing data. It would be great if Theorems 5.2 and 5.3 also came with similar quantitative bounds.

---

### Official Review · AnonReviewer1 · 2017-11-30

**Rating:** 8
**Confidence:** 4

**Review:**

The paper proposes an approach to train generators within a GAN framework, in the setting where one has access only to degraded / imperfect measurements of real samples, rather than the samples themselves. Broadly, the approach is to have a generator produce the "full" real data, pass it through a simulated model of the measurement process, and then train the discriminator to distinguish between these simulated measurements of generated samples, and true measurements of real samples. By this mechanism, the proposed method is able to train GANs to generate high-quality samples from only imperfect measurements.

The paper is largely well-written and well-motivated, the overall setup is interesting (I find the authors' practical use cases convincing---where one only has access to imperfect data in the first place), and the empirical results are convincing. The theoretical proofs do make strong assumptions (in particular, the fact that the true distribution must be uniquely constrained by its marginal along the measurement). However, in most theoretical analysis of GANs and neural networks in general, I view proofs as a means of gaining intuition rather than being strong guarantees---and to that end, I found the analysis in this paper to be informative.

I would make a  suggestions for possible further experimental analysis: it would be nice to see how robust the approach is to systematic mismatches between the true and modeled measurement functions (for instance, slight differences in the blur kernels, noise variance, etc.). Especially in the kind of settings the paper considers, I imagine it may sometimes also be hard to accurately model the measurement function of a device (or it may be necessary to use a computationally cheaper approximation for training). I think a study of how such mismatches affect the training procedure would be instructive (perhaps more so than some of the quantitative evaluation given that they at best only approximately measure sample quality).

---

> ### Comment · AnonReviewer1 · 2018-01-12
> **post-rebuttal**
>
> After reading the other reviews and responses, I retain a favorable opinion of the paper. The additional experiments are especially appreciated.

---

### Author Response · Authors · 2018-01-05
**Response to reviewers' comments**

We appreciate the reviewers’ helpful comments.

Reviewer1 and Reviewer2 both suggest further experimental analysis to evaluate the robustness of our approach to systematic mismatches between the true and modeled measurement functions. This is a great idea and towards this, we have performed the following experiment:

We consider the observed measurements in the block pixels model with the probability of blocking pixels (p*) = 0.5. We then attempt to use the AmbientGAN setup to learn a generative model without any knowledge of p*. We try several different values of p for the simulated measurements and plot inception score vs the assumed dropout probability p. Please see the plot in Appendix D of the updated pdf.

We observe that the inception score peaks at the true value and gradually drops on both sides. This suggests that using p only approximately equal to p* still yields a good generative model, indicating that the AmbientGAN setup is robust to systematic mismatches between the true and modeled measurement functions. It would be interesting to analyze the robustness properties further, both empirically and theoretically.

Reviewer2's comment also suggests attempting to estimate the parameters of the measurement function. This seems important in practical settings and we thank the reviewer for pointing this out. Going even further, one can also attempt to estimate the measurement function including its function form. We remark that distributional assumptions are necessary for any such procedure and it would be interesting to construct and analyze estimators under various settings. For instance, if we know that zero pixels are rare (e.g., the celebA dataset), then we can easily estimate the dropout probability by counting the number of zero pixels in the measurements. Further, since one cannot expect the estimation to be perfect, robustness, as alluded to above, is necessary. We are keen to explore these ideas further.

To answer Reviewer2’s question about getting a useful model, we attempted to use the GAN learned using our procedure for compressed sensing. Generative models have been shown to improve sensing over sparsity-based approaches (https://arxiv.org/abs/1703.03208). Through the following experiment, we show that a similar improvement is obtained using the GANs learned through the AmbientGAN approach.

We train an AmbientGAN with block pixels measurements on MNIST with p = 0.5. Using the learned generator, we follow the rest of the procedure in (https://arxiv.org/abs/1703.03208). Using their code (available at https://github.com/AshishBora/csgm) we can plot the reconstruction error vs the number of measurements, comparing Lasso with AmbientGAN. Please see the plot in Appendix D of the updated pdf; we find that the AmbientGAN model gives significant improvements for a wide range of measurements.

---

### Public Comment · ~Pouya_Samangouei1 · 2018-02-23
**Similar to Algorithm 2 of CSGAN**

Congratulations on the acceptance of this nice work as an oral paper in ICLR18.

Our paper in AAAI18 "Task-Aware Compressed Sensing with Generative Adversarial Networks" https://arxiv.org/abs/1802.01284 is closely related to this work and your initial ICML17 "Compressed Sensing Using Generative Models".

More specifically, Algorithm 2 of our paper without the discriminator for the original samples is very similar to AmbientGAN. Table 3 shows reconstruction results for this setting.  We have also shown that with an extra discriminator for the original samples, one can get better reconstructions in Tables 1 and 2. Besides the reconstruction part, we also have shown that one can regularize the z-space to be more discriminative for inference.

The code is also available at https://github.com/po0ya/csgan for further investigation.

---

### Decision · Program_Chairs · 2018-01-29
**ICLR 2018 Conference Acceptance Decision**

**Decision:**

Accept (Oral)

**Comment:**

All three reviewers were positive about the paper, finding it to be on an interesting topic and with broad applicability. The results were compelling and thus the paper is accepted.